# The human cerebellum is essential for modulating perceptual sensitivity based on temporal expectations

**Assaf Breska\*, Richard B Ivry**

Department of Psychology and Helen Wills Neuroscience Institute, University of California, Berkeley, Berkeley, United States

**Abstract** A functional benefit of attention is to proactively enhance perceptual sensitivity in space and time. Although attentional orienting has traditionally been associated with cortico-thalamic networks, recent evidence has shown that individuals with cerebellar degeneration (CD) show a reduced reaction time benefit from cues that enable temporal anticipation. The present study examined whether the cerebellum contributes to the proactive attentional modulation in time of perceptual sensitivity. We tested CD participants on a non-speeded, challenging perceptual discrimination task, asking if they benefit from temporal cues. Strikingly, the CD group showed no duration-specific perceptual sensitivity benefit when cued by repeated but aperiodic presentation of the target interval. In contrast, they performed similar to controls when cued by a rhythmic stream. This dissociation further specifies the functional domain of the cerebellum and establishes its role in the attentional adjustment of perceptual sensitivity in time in addition to its well-documented role in motor timing.

## Introduction

Adaptive behavior is facilitated by an attentional system that can proactively modify the state of perceptual systems. While the majority of research on the underlying mechanisms has focused on spatial orienting, the brain is also anticipatory in time (*Nobre and van Ede, 2018*). In temporal anticipation, the brain exploits various temporal regularities in the environment to predict the timing of upcoming sensory events. These temporal expectations can guide temporal orienting, the adjustment of attention in time to modulate perceptual sensitivity, for example, increasing it at expected compared to unexpected times (*Correa et al., 2005*; *Rohenkohl et al., 2012*; *Davranche et al., 2011*; *Denison et al., 2017*; *Samaha et al., 2015*; *Fernández et al., 2019*) (also referred to in the literature as 'temporal attention'). This form of anticipation has been often associated with left inferior parietal and ventral premotor cortices (*Davranche et al., 2011*; *Coull and Nobre, 1998*; *Bolger et al., 2014*), a network that overlaps with frontal-parietal components of the cortico-thalamic network traditionally implicated in attentional orienting in space (*Posner and Petersen, 1990*; *Corbetta and Shulman, 2002*; *Fiebelkorn and Kastner, 2020*).

Recently, we reported that individuals with cerebellar degeneration (CD) fail to exhibit reaction time benefits from temporal cues on a simple detection task (*Breska and Ivry, 2018*; *Breska and Ivry, 2020*). These findings implicate the cerebellum in temporal preparation, but do not enable to determine whether it has a role in attentional modulation of perceptual systems, in motor preparation, or in both. This is because in speeded detection tasks, reaction time benefits from temporal cues could result from adjustment of perceptual sensitivity and/or of motor preparation (*Rohenkohl et al., 2012*; *Sanabria et al., 2011*; *Morillon et al., 2016*; *van Ede et al., 2020*). Given the well-documented role of the cerebellum in precisely timed movement (*Ivry and Keele, 1989*; *Perrett et al., 1993*), the impairments we observed in the CD group may merely reflect a novel

**\*For correspondence:**
assaf.breska@berkeley.edu

manifestation of the cerebellar role in the temporal control of motor preparation (*Shalev et al., 2019*). However, in addition to its role in motor timing, the cerebellum could also be essential to attentional orienting in time to proactively modulate perception. Determining the functional domain of the cerebellum in temporal anticipation is critical to both models of attention and of cerebellar function.

To address this question, we compared the ability of individuals with CD and healthy controls to use temporal cues to benefit performance in a challenging non-speeded perceptual discrimination task, in which the benefits of attention are assumed to reflect modulation of perceptual sensitivity (*Correa et al., 2005*; *Davranche et al., 2011*; *Raymond et al., 1992*). Participants judged the orientation of a briefly presented visual target whose contrast was set to be near-threshold, making the task perceptually demanding. In each trial, a temporal cue indicated that the interval between the target and a preceding warning signal (WS) would either be 600 or 1000 ms (*Figure 1A*). Targets mostly appeared at the cued time (valid trials) and rarely at the uncued time (invalid trials). Critically, participants were only queried for a response after a substantial random delay following target offset. In tandem with our instructions that only emphasized accuracy, the inclusion of the delay eliminated the need for motor preparation. With these manipulations, we assume that the expected performance advantage on valid compared to invalid trials (validity effect; *Correa et al., 2005*;

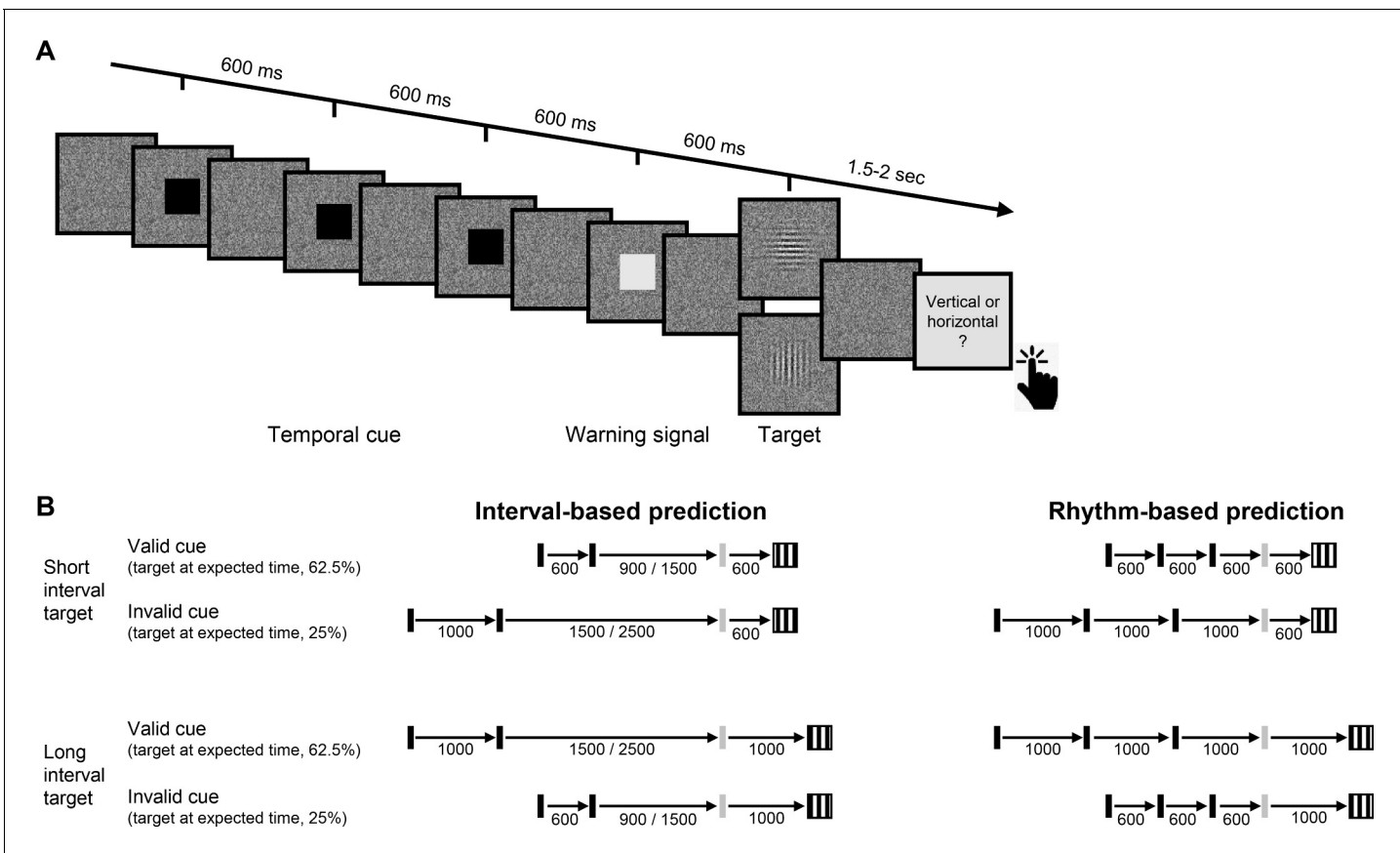

**Figure 1.** Experimental task. (A) Trial sequence, depicting a trial with the faster rhythmic temporal cue. Participants viewed a visual stream of black squares (temporal cue), followed by a white square (warning signal [WS]), and then a luminance-defined Gabor grating (target). Participants made a delayed, non-speeded judgment of the orientation of the grating, with the response elicited only after a variable time interval. Continuous dynamic masking was generated by a white noise visual stimulus mask that changed at 60 Hz. Whereas the black and white squares were highly visible when superimposed on the mask, the target was embedded in the noise, reducing its visibility. The target contrast was set on an individual basis using an adaptive procedure. (B) Temporal cue conditions. The target could appear at a short (600 ms, top) or long (1000 ms, bottom) interval after the WS. Left: Interval task. Two black squares were separated by either the short or long interval, with a random non-isochronous interval between the second black square and the WS. On valid trials (62.5%), this interval matched the WS-to-target interval; on invalid trials (25%), it matched the other interval (in the remaining 12.5% 'catch' trials there was no target and no response was required). Right: Rhythm task. Three black squares and the WS appeared with identical stimulus onset asynchrony (SOA) (short/long). Valid and invalid trials were as in the interval task.

*Coull and Nobre, 1998*; *Miniussi et al., 1999*; *Rohenkohl et al., 2014*; *Breska and Deouell, 2017*) will reflect temporally focused enhancement of perceptual sensitivity. If the cerebellum has a causal role in attention orienting in time at a non-motor level, the validity effect should be reduced in CD patients relative to controls.

In addition to manipulating the validity of the temporal cue, we also varied the manner in which this information was presented. In our previous work (*Breska and Ivry, 2018*; *Breska and Ivry, 2020*), we found that the cerebellar involvement was limited to when temporal anticipation required encoding and recalling an isolated interval, but not when temporal anticipation could be based on synchronization with a stream of rhythmic sensory events. We employ a similar manipulation in the present study, comparing perceptual benefits from interval and rhythmic temporal cues (*Figure 1B*).

## Results

Perceptual sensitivity was quantified using d', a measure derived from signal detection theory (*Green and Swets, 1966*). Temporal orienting should manifest as higher d' for valid compared to invalid trials. A four-way omnibus analysis of variance (ANOVA) revealed a significant validity effect across groups, tasks, and target intervals (main effect of cue validity: $F(1,23)=13.42$, $p=0.001$, $\eta_p^2=0.37$). Prior work has shown that in two-interval designs as implemented here, the validity effect is less robust or even absent for late targets (*van Ede et al., 2020*; *Miniussi et al., 1999*; *Correa et al., 2006*). Thus, we employed an a priori analysis plan that focused on trials in which the target appeared at the short interval (600 ms, 'early targets') to increase sensitivity for detecting attenuation of the validity effect (see Materials and methods). As expected, the validity effect was larger when the target appeared at the short interval (cue validity × interval interaction: $F(1,23)=3.71$, one-tailed $p=0.033$, $\eta_p^2=0.14$). Indeed, neither group showed a significant validity effect in either task for the late onset targets (all uncorrected p's > 0.09, *Figure 2—figure supplement 1*).

*Figure 2* presents d' values for short interval targets. A three-way ANOVA revealed a significant validity effect across tasks and groups ($F(1,23)=14.75$, $p=0.001$, $\eta_p^2=0.39$). Across tasks, the validity effect was larger in the control compared to the CD group (cue validity × group interaction, F(1,23)

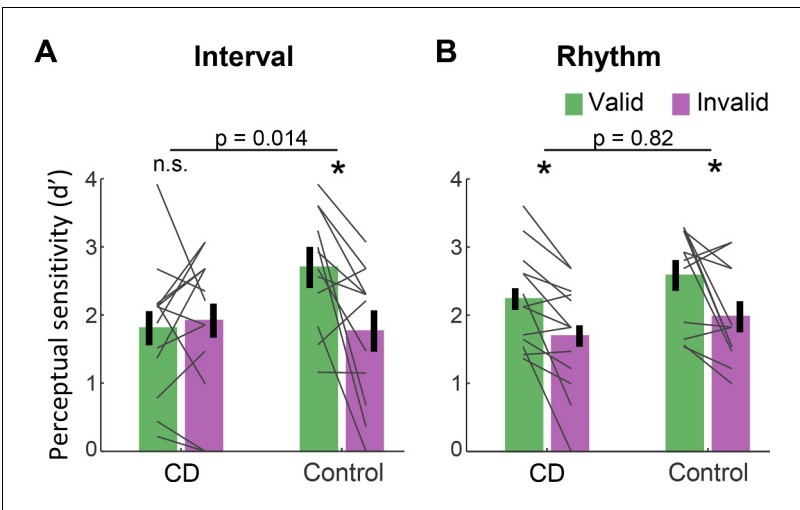

**Figure 2.** Absence of validity effect in individuals with cerebellar degeneration following interval-based, but not rhythm-based temporal cues. (**A**) Interval task. Mean d' for temporally expected (valid) and unexpected (invalid) short interval targets (600 ms after the WS) for the CD and control groups. Unlike the controls, the CD group showed no increase in d' when the target appeared at the expected time. (**B**) Rhythm task. Both groups show a similar increase in d' on valid trials. *p<0.05. In both **A** and **B**, error bars represent one standard error of the mean (SEM). Gray lines depict individual subject data.

The online version of this article includes the following figure supplement(s) for figure 2:

**Figure supplement 1.** Mean d' for temporally expected (valid) and unexpected (invalid) long interval targets (1000 ms after the warning signal [WS]).

=4.83, p=0.038, $\eta_p^2$=0.17), but this effect was qualified by a significant cue validity × group × task interaction (F(1,23)=4.76, p=0.04, $\eta_p^2$=0.17). We used a series of planned contrasts to evaluate the validity effect within each task. For the interval task (*Figure 2A*), the validity effect was significantly smaller in the CD group relative to the control group (two-way ANOVA, cue validity × group interaction: F(1,23)=7.11, p=0.014, $\eta_p^2$=0.24). The control group showed a higher mean d' for valid compared to invalid trials (t(11)=3.08, p=0.011, Cohen's d=0.89), while d' values in the CD group were not significantly different, and even numerically higher on invalid trials (t(12) = −0.43, p=0.67, BF$_{10}$=0.21, moderate evidence against a directional hypothesis of a validity effect). Thus, the CD group failed to show an enhancement in perceptual sensitivity from an interval-based temporal cue.

A different pattern was observed on the rhythm task (*Figure 2B*). Here, the magnitude of validity effect did not differ between the two groups (F(1,23)=0.053, p=0.82, $\eta_p^2$<0.01, BF$_{10}$=0.32). The control group again showed a validity effect (t(11)=2.67, p=0.022, Cohen's d=0.77). However, unlike in the interval task, the CD group also showed a validity effect (t(12)=3.45, p=0.005, Cohen's d=0.95). Thus, the ability to increase perceptual sensitivity at a specific time point based on a rhythm cue was preserved in the CD group. Furthermore, the presence of the three-way interaction implies that the attenuation of the validity effect in the CD group relative to controls was significantly larger in the interval task than in the rhythm task.

## Discussion

Our results provide compelling evidence that the integrity of the human cerebellum is necessary for proactive modulation of perceptual processing based on temporal expectations. The ability to use a temporal cue to enhance perceptual sensitivity at specific times was abolished in individuals with CD when prediction was based on an interval cue. In contrast, the CD group showed a comparable benefit to controls when temporal orienting was based on a rhythmic cue. Critically, the use of a non-speeded, challenging perceptual discrimination task argues strongly against the idea that the impairment is related to motor preparation processes. Rather, the results point to an essential context-specific role of the cerebellum in the temporal orienting of visual attention, adding to the substantial literature highlighting cerebellar involvement in a broad range of cognitive functions beyond the motor domain (*Sokolov et al., 2017*).

These findings point to the need for an expanded picture of the neuroanatomical network involved in attentional control of perceptual sensitivity. Attention research has traditionally emphasized cortico-thalamic networks, and in particular, dorsal and ventral fronto-parietal networks associated with top-down and stimulus-driven attention control, respectively (*Posner and Petersen, 1990*; *Corbetta and Shulman, 2002*; *Fiebelkorn and Kastner, 2020*). For attention orienting in the time domain, human imaging studies consistently reveal activations of the left inferior parietal cortex and the ventral premotor cortex (*Davranche et al., 2011*; *Coull and Nobre, 1998*; *Bolger et al., 2014*). Similarly, neural signatures of temporal anticipation identified in human electrophysiology studies have been localized to cortico-thalamic circuits (*Gómez et al., 2001*; *Praamstra, 2006*). In general, the cerebellum has not featured in this work, perhaps due to the difficulty in neuroimaging studies to separate the effects of temporal prediction on neural activations from the effects of other task parameters, or the difficulty of electrophysiology studies to measure cerebellar sources. One exception here is a study by *O'Reilly et al., 2008* who observed greater cerebellar activation for detecting violations of spatio-temporal predictions compared to spatial predictions, pointing to cerebellar involvement in temporal prediction (*O'Reilly et al., 2008*). The current findings demonstrate that the cerebellum has a causal role in temporal anticipation and establish its role in using temporal information to modulate attention for the perceptual analysis of visual features.

Our neuropsychological approach provides a more direct method to evaluate the contribution of the cerebellum to attention. Previous work, focusing on the spatial domain, has proven inconclusive. Some studies have found that individuals with focal cerebellar lesions show reduced benefits from cues indicating the spatial location of a forthcoming stimulus (*Townsend et al., 1999*; *Allen et al., 1997*). However, it has been proposed that these impairments may be motoric in nature, with the tasks confounding attentional demands with demands on eye movements and/or response preparation (*Ravizza and Ivry, 2001*; *Haarmeier and Thier, 2007*). These concerns do not apply to the current study given that the spatial aspects of the task were fixed and the motor requirements were

minimal, delayed until well after stimulus offset. We note that the current results do not address the question of whether the cerebellum, in addition to its role in temporal orienting of attention, is also involved in other spheres of attention.

The dissociation between the impairment in the interval task and the preserved performance in the rhythm task in the CD group has two important implications. First, a longstanding debate in the timing literature concerns whether temporal anticipation in a rhythmic context is mediated by rhythm-specific mechanisms (e.g., entrainment) or by the repeated operation of an interval-based mechanism (*Breska and Deouell, 2017*; *Haegens and Zion Golumbic, 2018*; *Drake and Botte, 1993*). Our results are at odds with the latter hypothesis given that the CD group failed to benefit from the interval cues yet showed normal benefits from the rhythm cues. Beat-based timing may rely on cortico-striatal circuits even in the absence of movement (*Cannon and Patel, 2021*), consistent with previous findings of impaired rhythm-based temporal prediction in Parkinson's disease (*Breska and Ivry, 2018*).

Second, the selective contribution of the cerebellum in interval-based but not rhythm-based contexts has been observed in multiple timing domains, including duration judgments, timed movement, and timed motor preparation (*Breska and Ivry, 2018*; *Breska and Ivry, 2020*; *Grube et al., 2010*; *Spencer et al., 2003*). Our findings extend this functional specificity to the attentional domain, pointing to a generalized role for the cerebellum in interval timing, at in the sub-second range. Notably, while inferences from single dissociations such as that observed in the present study can be limited by concerns about differences in task difficulty, this concern is alleviated by the comparable benefits observed in healthy controls from interval- and rhythm-based cues (*Breska and Ivry, 2018*; *Breska and Ivry, 2020*; *Breska and Deouell, 2017*) (also observed in current dataset: cue validity $\times$ task interaction within the control group, F(1,11)=0.86, p=0.37, $\eta_p^2$=0.07).

Computationally, how might the cerebellum contribute to the attentional control of perceptual sensitivity in time? Given the cerebellar involvement in interval-based timing across timing domains, an intuitive hypothesis is that the cerebellum is necessary for the temporal processing of isolated intervals (*Ivry and Keele, 1989*; *Ivry and Schlerf, 2008*). This could reflect a central role in one or both of two putative functional subcomponents in interval timing models (*Addyman et al., 2016*; *Gibbon et al., 1984*), the formation of a temporal representation or retrieval processes related to interval memory. By this view, predictive processing in non-cerebellar circuits (e.g., prefrontal cortex) relies on cerebellar interval timing capacities to parameterize the temporal dimension of the prediction. In rhythm-based orienting, an interval-based mechanism would not be required as the temporal parameters are contained within ongoing neural dynamics. However, a broader hypothesis is that the interval-based prediction itself is formed within the cerebellum, part of the cerebellar role in prediction in the motor domain and beyond (*Sokolov et al., 2017*; *Wolpert et al., 1998*; *Miall et al., 1993*). By this view, these cerebellar temporal predictions guide proactive modulation in non-cerebellar circuits according to task goals (e.g., to prepare perceptual or motor systems). In rhythm-based orienting, dedicated prediction mechanisms are not necessary due to the self-sustaining limit cycle properties of the putatively entrained oscillatory dynamics. Future work should aim to explore the separability of timing and prediction, identifying the cerebellar computations that provide the essential information for temporal orienting.

## Materials and methods

Key resources table

| Reagent type (species) or resource | Designation | Source or reference | Identifiers | Additional information |
|---|---|---|---|---|
| Software, algorithm | MATLAB 2019a | Mathworks | RRID:SCR_001622 | |
| Software, algorithm | R 3.6.3 | R project for statistical computing | RRID:SCR_001905 | |

### Participants

Fifteen individuals with CD and 14 age-matched neurotypical control individuals were recruited for the study. The data from two individuals from each group were discarded: One was unable to perform the task, two showed no convergence on the staircase procedure used to determine the perceptual threshold, and one asked to terminate the session prematurely. Thus, the final sample size

was 13 CD and 12 control participants. The sample size was determined using power calculations to allow 80% power to detect effects with Cohen's d = 0.8 (a conservative estimate, given the typical effect size of temporal cuing in prior studies: Cohen's d = 1–1.5; *Rohenkohl et al., 2012*; *Breska and Ivry, 2018*; *Rohenkohl et al., 2014*). All participants provided informed consent and were financially compensated for their participation. The study was approved by the Institutional Review Board at the University of California, Berkeley.

Participants in the CD group (9 females, 12 right-handed, mean age = 56.2 years, SD = 11.1) had been diagnosed with spinocerebellar ataxia, a slowly progressive adult-onset degenerative disorder in which the primary pathology involves atrophy of the cerebellum. We did not test patients who presented symptoms of multisystem atrophy. Eight individuals in the CD group had a specific genetic subtype (SCA3 = 2, SCA6 = 3, SCA17 = 1, SCA35 = 1, AOA2 = 1) and the other five individuals had CD of unknown/idiopathic etiology. All of the CD participants provided a medical history to verify the absence of other neurological conditions, and were evaluated at the time of testing with the Scale for the Assessment and Rating of Ataxia (SARA) (*Schmitz-Hübsch et al., 2006*). The mean SARA score was 13.5 (SD = 6.3). Control participants (8 females, 11 right-handed, mean age = 59.1, SD = 10.2) were recruited from the same age range as the CD group, and, based on self-reports, did not have a history of neurological or psychiatric disorders. The CD and control groups did not differ significantly in age (p=0.52).

All participants were prescreened for normal or corrected-to-normal vision and intact color vision. We also screened for professional musical training or recent amateur participation in musical activities (e.g., playing a musical instrument or singing in a choir), with the plan to exclude individuals with such experience (none did). All of the participants completed the Montreal Cognitive Assessment (MoCA) (*Nasreddine et al., 2005*) as a simple assessment of overall cognitive competence. Although we did not select participants to provide a match on this measure, there was no significant group difference (CD: mean = 26.7, control: mean = 27.5, p=0.32).

## Stimuli and task

For the experimental task, participants discriminated the orientation of a masked visual target, whose timing was cued on each trial (*Figure 1*). The target was a grayscale, luminance-defined sinusoidal Gabor gratings (size: 400 × 400 pixels, 11 × 11 cm, 10° visual angle; spatial frequency = 1 cycle/degree; Gaussian standard deviation = 2.5°) that was either oriented horizontally or vertically. The target was embedded in a dynamic, white noise mask. This mask was a square (size: 400 × 400 pixels) in which each pixel was randomly assigned a luminance value between 0.25 and 0.75 (with 0 and 1 being black, RGB: [0,0,0] and white, RGB: [255,255,255], respectively). The luminance value for each pixel in the mask was updated every 16.6 ms (once per monitor refresh cycle) throughout the trial. The contrast of the target relative to the background noise was adjusted for each participant (see below). Temporal cues were provided by black squares (size: 200 × 200 pixels, 5.5 × 5.5 cm, 5° visual angle). All stimuli were created in MATLAB (MathWorks, Natick, MA) and presented using the Psychtoolbox v.3.0 package for MATLAB (*Pelli, 1997*; *Brainard, 1997*). The stimuli were presented foveally on a gray background (RGB: [128,128,128]) on a 24-in monitor (resolution: 1920 × 1080 pixels, refresh rate: 60 Hz) at a viewing distance of ~65 cm.

The dynamic noise mask remained visible throughout the duration of the trial. The other stimuli were superimposed on this mask. The suprathreshold temporal cue involved the serial presentation of two or three black squares (100 ms duration each), with the first black square always appearing 750 ms after the onset of the dynamic mask. The black squares were followed by a suprathreshold white square, the WS (also 100 ms duration), which was followed in turn by the near-threshold target (50 ms duration) after either 600 ms (early target) or 1000 ms (late target). Note that given the screen refresh rate, the 50 ms target was successively embedded in three different masks.

The dynamic noise mask remained visible for 1700 ms after WS onset regardless of the target timing (1100 ms after early target onset, 700 ms after late target onset). After the termination of the mask, the screen was blank for a variable interval of 400–900 ms (randomly selected). At the end of this interval, a visual instruction appeared, requesting that the participant indicates the perceived orientation of the target (e.g., *vertical (press X) ? horizontal (press M)*). Responses were made with the X and M keyboard keys, assigned randomly for each participant and fixed for the entirety of the experiment. The participants were instructed that the discrimination would be difficult and, if uncertain, to make their 'best guess'. Note that the procedure involved a long delay between stimulus

offset and the response cue (1500–2000 ms for short interval targets and 1300–1600 ms for long interval targets). This long delay, coupled with the instructions, was included to eliminate the demands on motor preparation around the time of the target presentation and, as such, assure that observed benefits from the temporal cues arose from processes involved in perceptual discrimination.

Two types of sequences, tested in separate blocks, were used to provide temporal cues. In the interval task, the sequence consisted of two black squares, with a stimulus onset asynchrony (SOA) of either 600 ms (short cue) or 1000 ms (long cue). The SOA between the second black square and WS was randomly set on each trial to be either 1.5 or 2.5 times the cue interval on that trial (short cue trials: 900/1500 ms; long cue trials: 1500/2500 ms), strongly reducing any periodicity between the timing of the cue and target (*Breska and Deouell, 2017*). In the rhythm task, the sequence consisted of three black squares, presented periodically with an SOA of 600 ms (short cue, equivalent to 1.66 Hz) or 1000 ms (long cue, 1 Hz). The SOA between the third black square and WS was the same duration as the cue SOA for that trial. Thus, the WS fell on the 'beat' established by the temporal cues.

In both tasks, the SOA between the WS and target was either the same SOA as defined by the temporal cue (valid trial, 62.5% of trials) or the non-cued SOA (invalid trials, 25% of trials). This ratio was selected to incentivize the participant to use the temporal cues to facilitate performance on this challenging task. On the remaining 12.5% of the trials, no target was presented and the visual instruction screen did not appear. No response was required on these catch trials and the inter-trial interval commenced after the dynamic noise was terminated. We included these (no-response) catch trials to add uncertainty even when a target was not presented at the early target time, by this further increasing the difficulty of the task and exploring a possible validity effect in late targets.

## Procedure

Upon arrival, all participants provided consent, demographic information, and completed the MoCA. The CD participants also provided their clinical history and were evaluated with the SARA.

The experiment was conducted in a quiet, dimly lit room. The session began with a familiarization stage, in which participants performed four practice trials with 100% target contrast followed by four with 40% target contrast. The latter were included to demonstrate to the participants how difficult it could be to make a simple orientation judgment when the contrast of the target was similar to that of the mask.

Following this familiarization phase, we used an adaptive method to determine, on an individual basis, the target contrast level expected to produce discrimination accuracy of ~79% (descending staircase procedure, 3 down 1 up [*Levitt, 1971*], step size = 2%, 10 reversals). We opted to target 79% accuracy to provide sufficient room to detect improvement (to a ceiling of 100% performance) or impairment (to a floor of 50% performance). Importantly, for this adaptive procedure, only rhythmic temporal cues were used, and the target always appeared at the expected time (valid). Our reasoning here was that if the CD group were able to use the temporal cues to modulate perception, it is more likely to occur in this task (and have a similar threshold value as controls) given previous work showing that these individuals are not impaired in utilizing rhythmic temporal cues (*Breska and Ivry, 2018*). In this way, we would be positioned to ask if the CD group showed an impaired validity effect in the rhythm task as well as overall performance (valid and invalid trials) on the interval task. The contrast level identified from the adaptive procedure for a given individual was used in the main experiment for both tasks. Consistent with our expectation, the mean contrast level did not differ between groups (t(23)=0.83, p=0.41).

In the main experiment, participants preformed four blocks of each task, alternating between rhythm and interval blocks (eight blocks total). Each block consisted of 32 trials, 16 with the short temporal cue and 16 with the long temporal cue. Of these 16 trials, the target appeared at the cued time on 10 trials (valid), the uncued time on 4 trials (invalid), and did not appear on 2 trials (catch). When present, the target was horizontal on 50% of the trials and vertical on the other 50% of the trials. Short breaks were provided between each block.

To ensure that the target contrast fell in a range that would be optimal for detecting a validity effect, we calculated the averaged performance on valid trials across the two tasks after each pair of blocks. If the accuracy was higher than 99%, we reduced the target contrast for subsequent blocks by 4%, and if it was lower than 60%, we increased it by 4%. Block pairs in which performance was

above 99% or below 60% were not included in the d' analyses (four excluded blocks across all participants; exclusion had no impact on the statistical tests).

Prior to the first block for each task, the experimenter demonstrated the trial sequence and then conducted practice trials until the participant could describe how the cues were predictive of the onset time of the target. For subsequent blocks, the participant first completed two practice trials as a reminder of the format for the temporal cues in the forthcoming block. Participants received feedback on their performance after these practice trials (but not after any of the staircase or experimental trials).

### Statistical analysis

The data were analyzed using custom MATLAB scripts and R50. Following standard practices for data analysis in non-speeded, 2-AFC tasks, we quantified discrimination performance by calculating, on an individual basis, a d-prime (d') score separately for each combination of task, target interval and cue validity. These values were calculated by subtracting the z-score of the percentage of hits from the z-score of the percentage of false alarms (referring to vertical and horizontal categories as 'stimulus present' and 'stimulus absent', respectively, in classic signal detection terminology). As the 'hit' category was arbitrarily assigned to one orientation and the two orientations were equally probable, we did not calculate or analyze the criterion index. An increase in perceptual sensitivity due to temporal anticipation should be manifest as an increase in d' when the target appeared at the expected time compared to when it appeared at the unexpected time (validity effect).

Previous work indicates that validity effects from temporal cues in two-interval designs such as that used here are usually attenuated for late onset targets, either due to re-orienting of attention in time or foreperiod effects (*Correa et al., 2006*). As such, our a priori plan was to focus on trials with short interval targets to increase sensitivity for detecting attenuation of the validity effect. To confirm that this pattern was present in our data, we subjected d' values to an omnibus four-way mixed ANOVA with a between-subject factor group (CD/control), and within-subject factors task (interval/rhythm), target interval (early/late), and cue validity (valid/invalid), and tested the directional hypothesis of a larger effect of the cue validity factor for target appearing at early compared to late intervals using a one-tailed test. As expected (see Results), we observed a significant cue validity × target interval interaction in the expected direction, and post hoc comparisons showed that the validity effect was only significant in the early interval condition.

The d' values for short interval targets were analyzed using a mixed ANOVA with a between-subject factor group (CD/control), and within-subject factors task (interval/rhythm) and cue validity (valid/invalid). To assess the effect of cue validity within each group and task, we used within-subject t-tests. To compare the cue validity effect between groups within each task, we used a mixed ANOVA with factors group (CD/control) and cue validity (valid/invalid). Finally, to assess context specificity within the CD group, we performed an orthogonal contrast, comparing the cue validity effects between tasks using a repeated-measures ANOVA with factors task (interval/rhythm) and cue validity. In all analyses, effect sizes were estimated using Cohen's d and partial eta-squared ($\eta_p^2$).

## Acknowledgements

We thank Arohi Saxena for assistance in data collection.

## Additional information

### Competing interests

Richard B Ivry: Senior editor, *eLife*. The other author declares that no competing interests exist.

### Funding

| Funder | Grant reference number | Author |
|---|---|---|
| National Institutes of Health | NS092079 | Richard B Ivry |

| National Institutes of Health | NS116883 | Richard B Ivry |

The funders had no role in study design, data collection and interpretation, or the decision to submit the work for publication.

## Author contributions

Assaf Breska, Conceptualization, Data curation, Software, Formal analysis, Validation, Investigation, Visualization, Methodology, Writing - original draft, Project administration, Writing - review and editing; Richard B Ivry, Conceptualization, Resources, Supervision, Funding acquisition, Project administration, Writing - review and editing

## Author ORCIDs

Assaf Breska (iD) https://orcid.org/0000-0002-6233-073X
Richard B Ivry (iD) https://orcid.org/0000-0003-4728-5130

## Ethics

Human subjects: All participants provided informed consent to participate in the study and for the publication of de-identified data. The study was approved by the Institutional Review Board at the University of California, Berkeley (CPHS# 2016-02-8439).

## Decision letter and Author response

Decision letter https://doi.org/10.7554/eLife.66743.sa1
Author response https://doi.org/10.7554/eLife.66743.sa2

# Additional files

## Supplementary files

• Source data 1. Raw data.

• Source code 1. Preprocessing code (MATLAB).

• Source code 2. Statistical analysis code (R).

• Transparent reporting form

## Data availability

De-identified source data files for all figures and analyses in the article have been provided. Additional demographic information was not uploaded as it was not used in any analysis reported in the text, and can be provided upon request in personal communication with the corresponding author, without additional restrictions.

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
