## [Decision Letter]

**Acceptance summary:**

This study provides evidence that individuals with cerebellar degeneration show reduced effects of temporal expectation on perceptual discriminability with interval timing cues, but intact effects with rhythmic cues. The authors compare individuals with cerebellar degeneration to controls, and find a selective impairment of the individuals with cerebellar degeneration to use interval-based temporal predictions to facilitate visual discrimination, whereas rhythm-based performance benefits are spared. This study is of interest to psychologists and neuroscientists investigating prediction, perception, attention, and motor control, as it demonstrates a key role for the cerebellum in mediating the effects of interval-based temporal expectation on perception.

**Decision letter after peer review:**

Thank you for submitting your article "The human cerebellum is essential for modulating perceptual sensitivity based on temporal expectations" for consideration by *eLife*. Your article has been reviewed by 3 peer reviewers, and the evaluation has been overseen by a Reviewing Editor and Chris Baker as the Senior Editor. The following individual involved in review of your submission has agreed to reveal their identity: Freek van Ede (Reviewer #2).

The reviewers have discussed their reviews with one another, were enthusiastic about the study, and the Reviewing Editor has drafted this to help you prepare a revised submission.

Essential revisions:

The reviewers raised no major concerns about the study.

Minor Comments:

(1) At times the manuscript reads as if the cerebellum may be involved either in motor timing or in perceptual timing. I believe the authors position is that the cerebellum is likely involved in temporal anticipation generally, which could manifest in (benefits from) action planning, but also visual preparation – depending on nature of the task. This view is more clearly expressed in the discussion, but could potentially be made more explicit from the beginning. For example, the following sentence in the abstract ("While this deficit may reflect impairment in anticipatory motor preparation, it could also arise from cerebellar contribution to attentional modulation in time of perceptual sensitivity") may tempt the reader to expect this to be a study that tries to disambiguate whether the cerebellum does either motor preparation or visual preparation – while it may be involved in both.

(2) Given that the cerebellum is known to play a role in interval timing, is the interpretation that the impairments in temporal orienting in the interval task arise from difficulty with interval timing, or is the interpretation broader than that? Is there any chance there is data from these subjects on interval timing tasks, such that the authors could ask whether interval timing performance predicts temporal expectation effects?

(3) Making use of the interval information in the interval task requires not only timing the deployment of attention after the appropriate interval but also estimating the cue time interval and keeping that interval estimate in working memory until the target period. Is there any data that would speak to which of these temporal subtasks the cerebellum is involved in?

(4) A few more relevant references for effects of temporal attention on perceptual sensitivity: Denison et al. 2017 PBR, Fernandez et al. 2019 JOV, Samaha et al. 2015 PNAS

---

## [Author Response]

Minor Comments:(1) At times the manuscript reads as if the cerebellum may be involved either in motor timing or in perceptual timing. I believe the authors position is that the cerebellum is likely involved in temporal anticipation generally, which could manifest in (benefits from) action planning, but also visual preparation – depending on nature of the task. This view is more clearly expressed in the discussion, but could potentially be made more explicit from the beginning. For example, the following sentence in the abstract ("While this deficit may reflect impairment in anticipatory motor preparation, it could also arise from cerebellar contribution to attentional modulation in time of perceptual sensitivity") may tempt the reader to expect this to be a study that tries to disambiguate whether the cerebellum does either motor preparation or visual preparation – while it may be involved in both.

The reviewers are correct that, when considered in the context of other work, our view is that the cerebellum is involved in temporal anticipation in a general sense, and thus exploited for both action and perception. The key motivation for the current study is not to contrast “motor” vs “perception”, but to really put the perceptual component to test given that impairments on tasks using RT measures could be interpreted as reflecting problems with motor preparation. We do not mean to imply that these are mutually exclusive and have now revised relevant points in the Abstract and Introduction to make this point and clarify the objective of the study.

(2) Given that the cerebellum is known to play a role in interval timing, is the interpretation that the impairments in temporal orienting in the interval task arise from difficulty with interval timing, or is the interpretation broader than that? Is there any chance there is data from these subjects on interval timing tasks, such that the authors could ask whether interval timing performance predicts temporal expectation effects?

We agree with the reviewers that there is likely overlap between the cerebellar role in temporal expectation, an indirect probe of timing, and interval timing, a more direct probe of timing. We recognize the value of testing the same individuals on both sets of tasks and it is part of our future plans. We do not have interval timing data for the participants tested in this study. We do discuss this issue in the last paragraph of the Discussion, where we outline possible computational architectures – difficulty in interval timing, reflecting the cerebellar role in interval representation, or more broadly in prediction, in line with its putatively central role in motor control.

(3) Making use of the interval information in the interval task requires not only timing the deployment of attention after the appropriate interval but also estimating the cue time interval and keeping that interval estimate in working memory until the target period. Is there any data that would speak to which of these temporal subtasks the cerebellum is involved in?

As noted in our response to the previous comment, our findings cannot identify the specific functional component of interval timing that is impaired in the CD group—our focus here was on clarifying that the integrity of the cerebellum was important for perceptual gains associated with temporal attention. Following the reviewers’ comment we revised the Discussion to outline how time estimation and interval memory could be distinct (or overlapping) functional subcomponents that rely on the cerebellum.

(4) A few more relevant references for effects of temporal attention on perceptual sensitivity: Denison et al. 2017 PBR, Fernandez et al. 2019 JOV, Samaha et al. 2015 PNAS

We have added these references to the revised manuscript.